# Giant piezoresistive effect by optoelectronic coupling in a heterojunction

Thanh Nguyen[1], Toan Dinh[1], Abu Riduan Md Foisal[1], Hoang-Phuong Phan[1], Tuan-Khoa Nguyen[1], Nam-Trung Nguyen [1] & Dzung Viet Dao [1,2]

Enhancing the piezoresistive effect is crucial for improving the sensitivity of mechanical sensors. Herein, we report that the piezoresistive effect in a semiconductor heterojunction can be enormously enhanced via optoelectronic coupling. A lateral photovoltage, which is generated in the top material layer of a heterojunction under non-uniform illumination, can be coupled with an optimally tuned electric current to modulate the magnitude of the piezoresistive effect. We demonstrate a tuneable giant piezoresistive effect in a cubic silicon carbide/silicon heterojunction, resulting in an extraordinarily high gauge factor of approximately 58,000, which is the highest gauge factor reported for semiconductor-based mechanical sensors to date. This gauge factor is approximately 30,000 times greater than that of commercial metal strain gauges and more than 2,000 times greater than that of cubic silicon carbide. The phenomenon discovered can pave the way for the development of ultra-sensitive sensor technology.

[1] Queensland Micro- and Nanotechnology Centre, Griffith University, Brisbane, Queensland, Australia. [2] School of Engineering and Built Environment, Griffith University, Gold Coast, Queensland, Australia. Correspondence and requests for materials should be addressed to T.N. (email: thanh.nguyen11@griffithuni.edu.au) or to T.D. (email: toan.dinh@griffith.edu.au) or to D.V.D. (email: d.dao@griffith.edu.au)

Discovered by Smith in 1954[1], the piezoresistive effect has been utilized as a major mechanical sensing technology[2]. Piezoresistive sensitivity refers to the fractional change in resistance under applied strain, known as the gauge factor (GF). This sensing technology can be found in a wide range of applications, such as strain, force, pressure, and tactile sensors[3] and accelerometers[4,5]. The advantages of this sensing concept include, but are not limited to, low power consumption, simple readout circuits and miniaturization capability[5]. However, the performance of the piezoresistive effect depends upon the carrier mobility, which is fundamentally limited by the nature of the piezoresistive materials.

The enhancement of the piezoresistive effect has been of great interest for developing ultra-sensitive sensing devices. The conventional strategy focuses on the arrangement of piezoresistors in optimal crystal orientations[6,7]. For example, the longitudinal piezoresistive coefficient of p-type silicon (100) is ~$6.6 \times 10^{-11}$ $Pa^{-1}$ in the [100] direction, while its value increases to $71.8 \times 10^{-11}$ $Pa^{-1}$ in the [110] direction[1,6]. The GF of single crystalline p-type cubic silicon carbide (3C-SiC) is 5.0 and 30.3 in the [100] and [110] orientation, respectively[8]. A significant improvement in piezoresistive sensitivity has also been demonstrated using optimal-doping concentrations[6,9]. In terms of material choice, metal strain gauges have been commercialized and are widely employed in industry, research and daily life. However, the piezoresistive effect in metals is fundamentally based on a geometry change under applied strain, resulting in a low GF that is typically less than 2[2]. Semiconductors such as silicon (Si) and silicon carbide (SiC) have emerged as suitable materials for strain sensing because of their relatively high GF of up to 200 in Si[2] and 30 in SiC[8,10]. While the strain-induced geometry change can be neglected in these semiconductors, the carrier mobility governs the piezoresistive performance.

Interestingly, a significant enhancement of the piezoresistive effect can be achieved by scaling down piezoresistors to the nanometer-scale owing to advanced nanofabrication techniques. At the nanoscale level, the charge mobility and surface-to-volume ratio considerably increase, resulting in a significant improvement in the strain sensitivity[11–13]. For instance, a large piezoresistive effect in top-down fabricated silicon nanowires (SiNWs) has been observed with a longitudinal piezoresistive coefficient of up to $-3550 \times 10^{-11}$ $Pa^{-1}$, which is almost 38 times greater than that of bulk Si[13]. However, the reliability of a large piezoresistive effect on the nanoscale is still controversial[14–17].

More recently, coupling piezoresistivity with other physical effects, such as piezoelectricity, has emerged as an advanced and promising approach to boost piezoresistivity. As such, the strain-modulated electric potential in piezoelectric materials, known as piezotronics, can be used to control or tune the transport of charge carriers. By utilizing strain-induced piezoelectric polarization charges at the local junction of zinc oxide (ZnO) nanowires to modify their energy band structures, Jun Zhou et al.[18] successfully demonstrated an increase in the GF from 300 to 1250, when the strain increased from 0.2% to 1%. Additionally, an electrically controlled giant piezoresistive effect in SiNWs has been reported with a GF of up to 5000 by employing an electrical bias to manipulate the charge carrier concentration[19]. Coupling of multiple physical effects in nanostructures has also been employed to modulate electrical transport in logic circuits[20], enhance the sensitivity and detection resolution of bio/chemical sensors[21–23], and improve the photovoltaic performance of solar cells[24]. An enhancement of up to 76% of the output voltage has been revealed in solar cells by modulating the interfacial charge transfer in indium phosphide/zinc oxide (InP/ZnO) heterojunctions by applying temperature gradients across the device[24].

In the present work, we report the discovery of a giant piezoresistive effect in semiconductor heterojunctions by coupling the photoexcitation of the charge carriers, the strain modification of the carrier mobility and the electric field modulation of the carrier energy. Visible light is utilized to illuminate the top layer material of the heterojunction structure, in which the sensing element is non-uniformly illuminated by a vertical visible light. This illumination generates a lateral photovoltage that is counteracted by an externally controlled electric field to tremendously modulate the magnitude of the piezoresistive effect. As a proof of principle, we employ a 3C-SiC nanofilm grown on a Si substrate to form a 3C-SiC/Si heterojunction. Under visible light illumination, a stable GF value of the SiC/Si heterojunction as high as ~58,000 is achieved, which is the highest value ever reported for semiconductor piezoresistive sensors. The piezoresistive effect in the SiC nanofilm is utilized to detect mechanical stress or strain, while its sensitivity is boosted by optimally and simultaneously regulating both the lateral photovoltage and the tuning current. While heavily doped p-type 3C-SiC/p-type Si ($p^+$-3C-SiC/p-Si) is used in this work, our method could also be extended to enhance the sensitivity of other materials and smart structures that have simultaneous photovoltaic and piezoresistive properties. Thus, our findings can open a new era for the development of ultra-sensitive mechanical sensors.

## Results

**Enhancement of the piezoresistive effect**. We demonstrated an unprecedentedly large piezoresistive effect in $p^+$-3C-SiC/p-Si heterojunctions using a bending method. The carrier concentrations in the 3C-SiC nanofilm and Si substrate were $5 \times 10^{18}$ and $5 \times 10^{14}$ $cm^{-3}$, respectively. The light intensity was 19,000 lx, while three different strains of 225, 451, and 677 ppm were induced in the material (Fig. 1). In our experiments, we supplied an optimally controlled tuning current and simultaneously measured the output voltage.

We supplied a constant tuning current of 29.75 μA and light with an intensity of 19,000 lx. Strain was periodically applied (i.e., Load ON) and released (i.e., Load OFF). The fractional change in

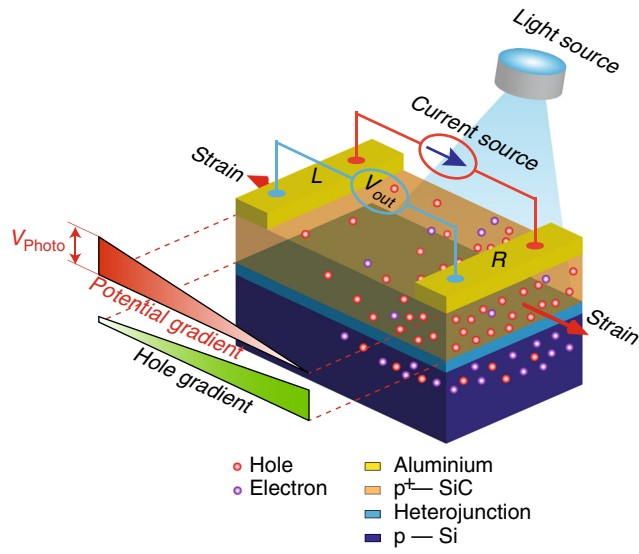

**Fig. 1** The sensitivity enhancement by optoelectronic coupling. The piezoresistive effect in the cubic silicon carbide/silicon heterojunction was modulated by non-uniform visible light illumination on the surface of the sensing element coupled with an optimally controlled tuning current

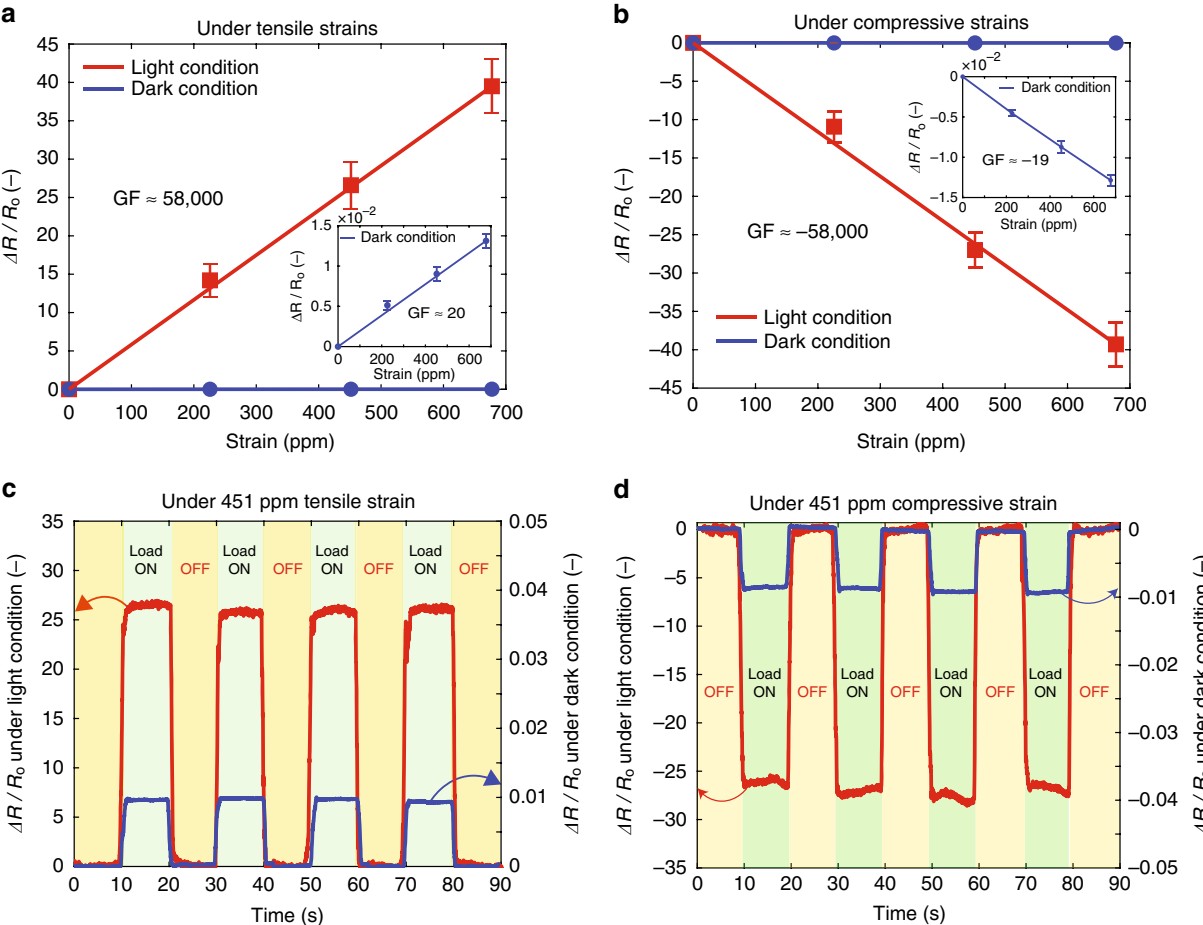

**Fig. 2** Enhancement of the piezoresistive effect. An unprecedented high gauge factor (GF) was achieved by simultaneously utilizing the lateral photovoltage and tuning the electric current to modulate the performance of the piezoresistive effect under tensile strain **a** and compressive strain **b**. The linearity of the sample applied in this method is excellent. Due to our discovery, a GF as high as 58,000 was achieved under tensile and compressive strain. **c** Repeatability of the fractional change in resistance under dark and light conditions as 451 ppm tensile strain was periodically turned ON and OFF. **d** Repeatability of the fractional change in resistance under dark and light conditions as the 451 ppm compressive strain was periodically turned ON and OFF

the resistance ($\Delta R/R_0$) was calculated as follows:

$$\frac{\Delta R}{R_0} = \frac{R - R_0}{R_0} = \frac{\frac{V}{I} - \frac{V_0}{I}}{\frac{V_0}{I}} = \frac{V - V_0}{V_0} = \frac{\Delta V}{V_0} \quad (1)$$

where the strain-free resistance $R_0$ is calculated by $R_0 = V_0/I$, $V_0$ is the voltage measured between the two electrodes under strain-free conditions, and $I$ is the supplied tuning current flowing between the two electrodes, which was kept constant throughout the measurement. When a strain or stress is applied, the resistance $R$ will change due to the piezoresistive effect. The value of resistance is calculated by $R = V/I$, where $V$ is the voltage measured between two electrodes under stress/strain application. As shown in Fig. 2a, b, the fractional changes in the resistance $\Delta R/R_0$ linearly increased with the increases in the tensile and compressive strain, which is desirable for high-performance strain-sensing applications. Figure 2c compares the fractional changes in the resistance $\Delta R/R_0$ between dark and light conditions under 451 ppm tensile strain, while Fig. 2d shows $\Delta R/R_0$ under 451 ppm compressive strain. Under a 451 ppm tensile strain, the $\Delta R/R_0$ value increased ~2950 times from 0.009 in the dark condition to 26.6 under light illumination, and this trend was similar under the compressive strain (the $\Delta R/R_0$ value increased from −0.0087 to −27 under the 451 ppm compressive strain). These results indicate a giant enhancement in the piezoresistive effect under light conditions. This tremendous

enhancement was confirmed under other applied strains as well (Supplementary Fig. 1). The piezoresistive sensitivity is characterized by the GF, which is defined as the fractional resistance change in response to the applied strain:

$$\text{GF} = \frac{\Delta R}{R_0} \times \frac{1}{\epsilon} = \frac{\Delta V}{V_0} \times \frac{1}{\epsilon} \quad (2)$$

where $\epsilon$ is the applied strain, which is detailed in Supplementary Table 1. Under the tensile strain, the GF was found to be 20 in the absence of light (the inset in Fig. 2a) and increased to ~58,000 under light illumination (Fig. 2a), which is the highest strain sensitivity ever reported. Moreover, under compressive strain, the change in the resistance or GF was similar to that observed under tensile strain but opposite in sign (Fig. 2b). In addition, the signal to noise ratio (SNR) under light condition increased significantly in comparison with that under dark condition.

**Tuning current in optoelectronic coupling.** The dependence of the piezoresistive effect on the tuning current under the 451 ppm tensile strain and the illumination condition of 19,000 lx intensity is depicted in Fig. 3. Figure 3a shows the change in the GF as the tuning current increased from 15 to 45 μA, while Fig. 3b–d illustrate the magnified graphs of the GF value versus the tuning current in three distinguished current ranges. The GF increased from approximately −16 to −1800 as the current increased from

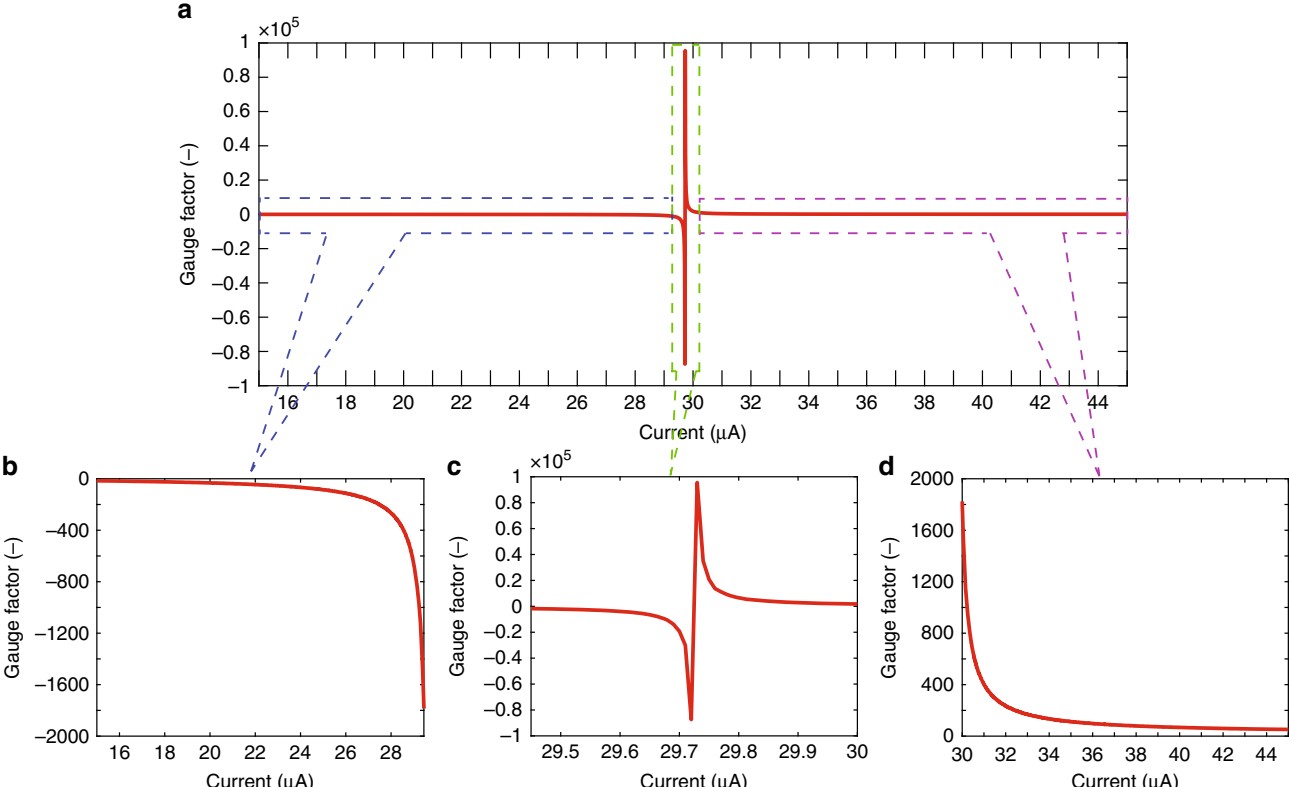

**Fig. 3** The role of the tuning current in the enhancement of the piezoresistive effect. **a** Characteristics of GF with respect to the supplied tuning current. Three ranges of tuning current are magnified in **b**, the negative GF range; in **c**, the optimal range; and in **d**, the positive GF range. Under the same light condition, the sensitivity (i.e., GF) changed significantly versus the supplied current. Under a light intensity of 19,000 lx, as the supplied current was swept from 15 to 45 μA, the GF increased from approximately −16 to a maximum of ~95,500, and then decreased to about ~50

15 to 29.45 μA (Fig. 3b), while it decreased from 1800 to ~50 for the high current ranging from 30 to 45 μA (Fig. 3d). This difference was attributed to the dominance of the photomodulated potential over the injected potential (i.e., tuning current). The compensation of these two potentials in the current range of from 29.45 to 30 μA led to a change in the GF sign and the ultra-high absolute GF values (Fig. 3c). As the strain-free voltage $V_0$ was relatively small due to the potential compensation, the modulation of the charge mobility under strain resulted in a significant change in the measured voltage, resulting in an ultra-high GF. It was observed that a higher GF can be achieved as the magnitude of the tuning current became closer to that of the photocurrent. For instance, under an incident light intensity of 19,000 lx, the maximum GF observed was as high as ~95,500. However, as the tuning current became closer to the photocurrent, the GF was more variable due to an inevitable slight variation in the photocurrent. Therefore, to achieve a higher stable sensitivity, the tuning current should be controlled and maintained as close as possible to the magnitude of the photocurrent but also far enough to retain stability. In our experiment, we achieved a stable GF of ~58,000 when the tuning current was help constant at 29.75 μA under an illumination of 19,000 lx from a stable visible light source.

As such, the significant enhancement in the piezoresistive effect by optoelectronic coupling in 3C-SiC/Si heterojunctions is a combination of two key elements: light illumination and tuning current. This enhancement was first attributed to the photo-generated electrical potential in the 3C-SiC film with non-uniform illumination of visible light, which was indicated by the lateral photovoltage and/or photocurrent. The magnitudes of the photovoltage and the photocurrent can be manipulated by

parameters, such as light intensity, light position, or light wavelength. The lateral photovoltage, for example, was measured to be approximately −9 mV under a light intensity of 19,000 lx (Fig. 4a), and the value of the photocurrent was ~29.7 μA (Fig. 4b). The magnitudes of the generated photovoltage and photocurrent can be changed by changing the light position. For instance, using the same previous light, we gradually adjusted the light beam position from the left (L) electrode to the right (R) electrode (Supplementary Fig. 2). The measured voltage decreased from a large positive value (e.g., 9 mV) at electrode L to zero at the center of the device and then increased to a large negative value at electrode R (e.g., −9 mV). The underlying physics behind the generation of the photocurrent and photo-voltage on 3C-SiC can be explained according to the lateral photoeffect[25]. Figure 5a shows the photon excitation of electron–hole pairs (EHPs) in the 3C-SiC/Si platform under light illumination. As such, photogenerated charge carriers have a high concentration at electrode R close to the light source. The formation of the gradient of charge carriers is discussed as follows.

**Formation of the gradient of charge carriers**. When heavily doped p-type 3C-SiC and p-type Si are brought together, holes diffuse from the 3C-SiC film into the Si substrate to decrease the hole gradient, leaving behind negative charges in the SiC layer near the interface of the heterojunction. In contrast, electrons in Si, as minor carriers, migrate into SiC and create a positive charge layer on the Si side. The migration of electrons and holes forms a depletion region (space charge region) and a built-in electric field $E_0$, which bends the conduction band and valence band in the

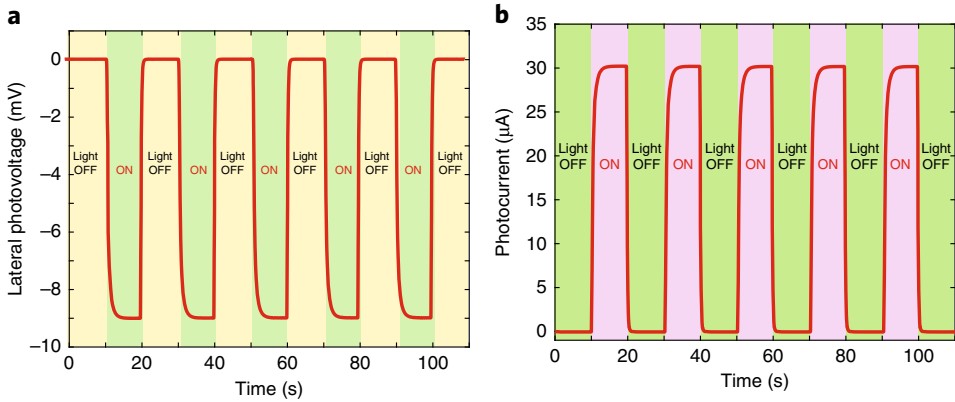

**Fig. 4** Generated photovoltage and photocurrent. **a** Under light illumination of 19,000 lx, a lateral photovoltage of approximately −9.0 mV was generated between the two electrodes. **b** The photocurrent under light illumination of 19,000 lx. As the light was turned OFF, the photocurrent was 0 μA, and this value was ~29.7 μA when light with an intensity of 19,000 lx was ON

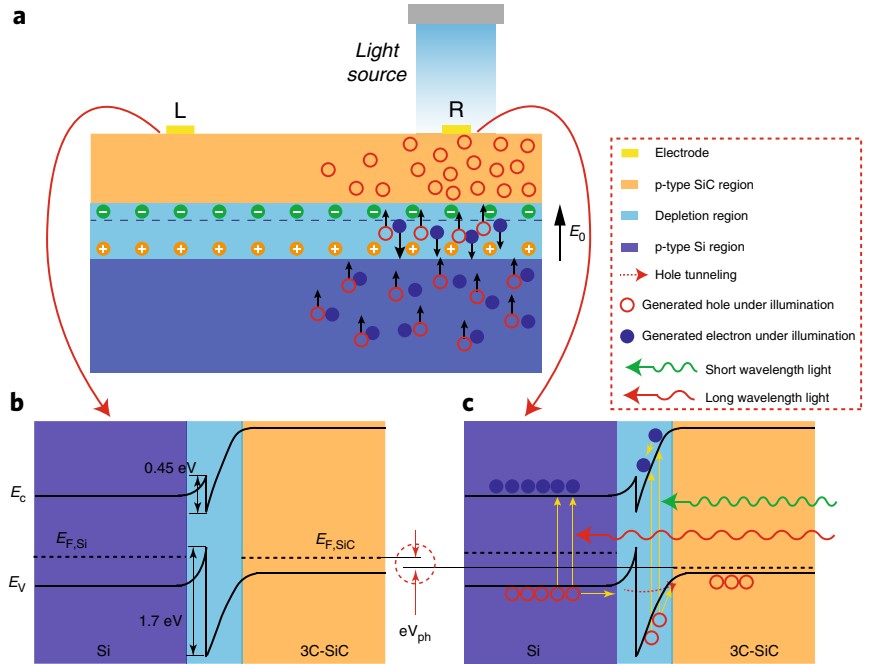

**Fig. 5** Formation mechanism of the lateral photovoltage. **a** Electron–hole pair generation and separation. Under visible light, electron–hole pairs are generated in the depletion region and Si substrate. Then, the generated EHPs in the depletion region are separated by an internal electric field, and the generated holes in the Si substrate tunnel to the 3C-SiC thin film. Schematic band diagrams of the heterojunction at the electrode L area without illumination **b** and at the electrode R area with illumination **c**. Under non-uniform illumination, photons are injected into the electrode R area rather than into the electrode L area, resulting in holes generated and injected in this area. Consequently, there is a gradient of the hole concentration in the 3C-SiC layer

depletion region. It is worth noting that the depletion region extends primarily into the Si substrate (Fig. 5a) because the carrier concentration in the Si substrate ($5 \times 10^{14}$ cm$^{-3}$) is much lower than that in the SiC thin film ($5 \times 10^{18}$ cm$^{-3}$). As shown in Fig. 5b, c, there are energy offsets of 0.45 and 1.7 eV for the conduction band and valence band, respectively, between 3C-SiC and Si[26]. Figure 5b, c present the band energy diagrams of SiC/Si under light illumination at electrode L and R areas, respectively. Owing to the visible-blind property of SiC, photons were only absorbed in the depletion region and the Si layer, where EHPs were generated. The generated EHPs in the depletion region were separated by the internal electric field $E_0$. Consequently, photogenerated holes in the depletion region moved to the SiC film and increased its electrical conductivity. We hypothesized that the photogenerated holes in Si also moved towards SiC by the

tunneling mechanism. Under non-uniform illumination, the majority of photons migrated into the electrode R area rather than into the electrode L area, resulting in holes were injected into this area. Consequently, there was a potential gradient in the hole concentration from electrode R to electrode L, which resulted in a difference in the electric potential described as $eV_{ph} = E_{F,SiC@R} - E_{F,SiC@L}$, where $e$ is the elementary charge, $E_{F,SiC}$ is the Fermi energy level, and $V_{ph}$ is the generated lateral photovoltage. When the external circuit was shorted, the only current in the circuit was the photocurrent ($I_{photo}$).

**Hole redistribution**. The potential gradient of the hole concentration from R to L can also be represented in the $E–k$ (energy–momentum) diagrams in Fig. 6. Under inhomogeneous illumination of light (Fig. 6a), the difference in the

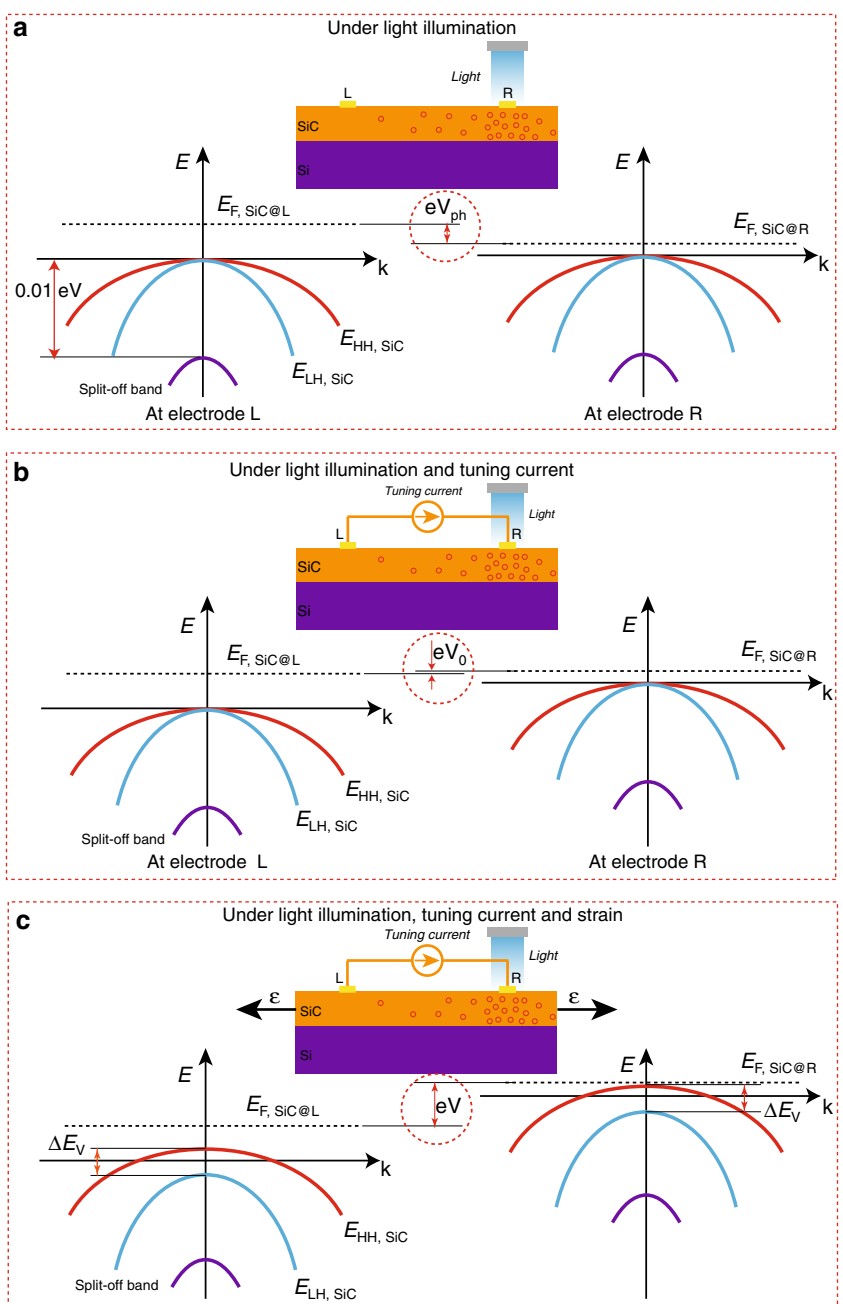

**Fig. 6** *E–k* (energy–momentum) characteristics of 3C-SiC nanofilms under different conditions. **a** *E–k* characteristics at electrode L and electrode R under illumination. **b** Under illumination and tuning current. **c** Under illumination, tuning current and applied strain

photogenerated hole concentration at R and L resulted in a difference in the Fermi levels in the SiC thin film in the two electrode regions. When a bias current $j$ with a positive terminal at electrode R and a negative terminal at electrode L is applied, the 3C-SiC band energy was bent upwards from electrode L to electrode R (Fig. 6b). This bias current created an electric field $E_b$

$$E_b = \int_L^R j \cdot \frac{1}{\sigma} \mathrm{d}x \qquad (3)$$

where $\sigma$ and $x$ are the conductivity of SiC and the distance from electrode L, respectively. This electric field $E_b$ offset the lateral photogenerated electric field $E_{ph} = eV_{ph}$, resulting in a relatively small voltage $V_0$ between the two electrodes. Particularly, under a light intensity of 19,000 lx, a bias current of 29.75 μA almost canceled out the lateral photovoltage, resulting in a nearly zero

voltage ($V_0 \approx 0$) (Fig. 6b). Figure 6c shows the change in the band diagram at the electrodes under uniaxial tensile strain. The energy sub-band of heavy holes (HHs) was shifted up to a lower energy level, while the energy sub-band of light holes (LHs) moved down to a higher energy level[27,28]. As a consequence, there was an increase in the HH concentration and a decrease in the LH concentration, while the total concentration of holes was hypothesized to be unchanged due to the high doping concentration. It should be noted that as HHs have a higher effective mass than LHs, the increase in the HH concentration and the decrease in the LH concentration caused an increase in the total of the effective mass. Consequently, the mobility of the holes was reduced, which diminished the conductivity $\sigma$ or increased resistance. As a result, the bias current generated a high electric field $E$ and a high measured voltage $V$. The significant difference between the

voltage $V$ under light illumination coupled with the applied strain (Fig. 6c) with respect to the nearly zero voltage $V_0$ under the strain-free state (Fig. 6b) led to the giant piezoresistive effect in the SiC nanofilms. Furthermore, in principle, it is possible to tune $V_0$ towards zero by regulating the illumination conditions and tuning the current to achieve a desirable high sensitivity to strain.

## Discussion

We discovered a giant GF of 58,000 in 3C-SiC/Si heterojunctions under optoelectronic coupling. This is the highest GF reported to date, which is ~30,000 times greater than the GF of commercial metal strain gauges and more than 2000 times higher than that of 3C-SiC under dark conditions. We analyzed three key parameters that contributed to this tuneable giant piezoresistive effect. First, non-uniform illumination created a gradient of carrier concentration within the 3C-SiC nanofilm, generating a lateral photovoltage in this layer. Second, the tuning current was introduced to reduce the difference in the Fermi energy levels of 3C-SiC at the two electrodes (L and R). Depending on the value of the lateral photovoltage, the optimal tuning current can have different values. Third, mechanical stress or strain caused shifts in the valance subbands (light hole and heavy hole), leading to the redistribution of charge carriers among these bands and changing the mobility and electrical conductivity of the material. The discovery of this optoelectronic coupling phenomenon in semiconductor heterojunctions has great potential to open a new era of ultra-sensitive sensor technology.

## Methods

**Growth of 3C-SiC on a Si substrate**. Single crystalline cubic silicon carbide (3C-SiC) was grown on a single crystalline Si substrate by low pressure chemical vapor deposition in a 1000 °C reactor. Ultra-pure silane and acetylene were used as precursor materials to provide Si and C in the 3C-SiC growth process. Heavily doped 3C-SiC was then formed by doping aluminum atoms from the $(CH_3)_3Al$ (trimethylaluminium) precursor compound in the in situ growth process. The characteristics of single crystalline 3C-SiC on a single crystalline Si substrate are shown in Supplementary Fig. 3[14]. Supplementary Figure 3a (the selected area electron diffraction image) and 3b (X-ray diffraction analysis) indicate that single crystalline 3C-SiC was epitaxially grown on a single crystalline Si substrate. The transmission electron microscopy image in Supplementary Fig. 3c confirms the crystalline properties of the SiC film. The thickness of the 3C-SiC layer measured by NANOMETRICS Nano-Spec-based measurements was 300 nm with the tolerance across the wafer within ±2 nm. The carrier concentrations in the 3C-SiC layer and single crystalline Si substrate were $5 \times 10^{18}$ and $5 \times 10^{14}\,cm^{-3}$, respectively, as determined by the hot probe and Hall effect techniques.

**Sample fabrication**. To demonstrate the piezoresistive effect of optoelectronic coupling in heterojunctions, cantilevers were fabricated according to the design shown in Supplementary Fig. 4. The length, width, and thickness of the cantilevers were 32, 5, and 0.63 mm, respectively. The distance from the free end of the cantilever to the centre of the piezoresistor was 25 mm. The dimensions of the piezoresistor were 0.5 mm × 2.5 mm, while those of the electrodes were 0.8 mm × 2.5 mm. Five cantilevers were fabricated following the process presented in Supplementary Fig. 5. After the growth process, an aluminum layer was deposited on top of 3C-SiC by a sputtering process. Then, a photoresist layer was coated on the surface of aluminum by a spin-coating technique at a spinning speed of 3500 rpm, and the photoresist layer was baked at 110 °C for 100 s. Next, the wafer was exposed to ultraviolet light to pattern the shape of the electrodes. The aluminum electrodes were formed through an aluminum wet etching process, which was followed by a final dicing process to form the cantilevers. A root mean square roughness of the top surface of the cantilever estimated by atomic force microscopy measurements was smaller than 15 nm. As shown in Supplementary Fig. 6, the $I–V$ characteristics were linear under both dark and light conditions, which confirmed that Ohmic contact formed between aluminum and 3C-SiC.

**Optoelectronic coupling characterization**. To characterize the optoelectronic coupling effect, five cantilevers were tested under the same conditions with the same procedure. As shown in Supplementary Fig. 7, the cantilevers were mounted on the chuck of the EP 6 CascadeMicrotech probe system. The sample position could be accurately adjusted. The samples were illuminated by vertical visible light from the fibre-optic illuminator used in the EP 6 CascadeMicrotech probe system, and the light beam position could also be precisely controlled using a precision XYZ stage. The light intensity measured using a digital lux meter was 19,000 lx.

The tensile and compressive strains on the SiC devices were induced using a cantilever bending experiment. Three different weights of 50, 100, and 150 g were hung on the free end of the cantilever to induce strain in the sensing element. The strain calculation is detailed in Supplementary Note 1. We controlled the tuning electric current and simultaneously measured the voltage between the two electrodes using a Keithley 2450 SourceMeter.

## Data availability

All relevant data of this work is available from the corresponding author upon reasonable request.

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

## Acknowledgements

The 3C-SiC material was developed and supplied by Leonie Hold and Alan Iacopi of the Queensland Microtechnology Facility, part of the Queensland node—Griffith—of the Australian National Fabrication Facility. A company established under the National Collaborative Research Infrastructure Strategy to provide nano and microfabrication facilities for Australia's researchers. The epitaxial SiC deposition was developed as part of Griffith Universities Joint Development Agreement with SPT Microtechnology, the manufacturer of the Epiflx production reactor. This work has been partially supported by Australian Research Council grants LP150100153 and LP160101553. T.D. is grateful for the support from Griffith University/Simon Fraser University Collaborative Travel Grant 2017 and Griffith University New Research Grant 2019.

## Author contributions

T.N., T.D. and A.R.F. designed and carried out the experiments. T.D., H.-P.P. and T.-K.N. fabricated the samples. T.N. and T.D. analyzed the data. T.N., T.D., N.-T.N. and D.V.D. discussed the results. T.N., T.D. and D.V.D. co-wrote the paper. All authors commented on the manuscript.

## Additional information

**Competing interests:** The authors declare no competing interests.

