## [Peer Review File · Nature Communications]

Reviewers' comments:

Reviewer #1 (Remarks to the Author):

The paper titled, "Towards infinite piezoresistance by optoelectronic coupling" reports the modulation of the piezoresistive effect by the use of light shining on the device thereby modulating the piezoresistance. It is a nice and clear paper that shows potential for ultra-sensitive strain sensors. Adequate information has been provided to permit a clear understanding of the phenomena and the methodology involved. However, the choice of the word, "infinite" in the title is a bit puzzling and needs some explanation.

It would be important to include a reference to Smith's 1954 paper. After mentioning that Smith discovered piezo-resistance in 1954, there is no reference to this seminal work. Please add a reference to the paper by Smith from 1954.

There are numerous errors of grammar. These need to be corrected in a paper submitted to a journal with this impact.

e.g. Abstract 2nd line - the word "giantly" is out of place. It should be replaced by "greatly" or some other alternate.

page 1 - 4th paragraph 16th line - "An enhancement of up to 76% has been revealed in solar cells"

There are numerous such errors all throughout the paper that must be corrected.

Does the doping concentration, crystalline quality or defect density of the grown SiC layer influence the gauge factor?

What influence, if any, does the wavelength of the shining light have on the data?

How reproducible is the data? How many cantilevers have been studied and does the data given in the paper reflect best values or average values for various cantilevers?

Kindly address these issues as well as revise the whole paper to correct the grammatical mistakes, etc

Reviewer #2 (Remarks to the Author):

This paper reports the photo-assisted enhancement of piezoresistive effect based on a SiC-Si heterostructure. Reported gauge factor of 58000 is highest value among reported values, but some details are missing, as pointed out below.

1. Meaning of V_0 is not clear. How to evaluate the resistance is not described.
2. Actual piezoresistive resistance changes in application of strain is not described. In this experiment, both tuning current and photocurrent are flowing in the resistive element. Observed voltage across the resistor can be very small if the tuning current and photocurrent are almost valanced. In that situation, the large gauge factors are possibly observed. Thus, the large gauge factor seems be a kind of trick. Actual resistance variation seems to be very small, than observed one. How to observe the actual resistance should be clear, and the resistance variation should be plotted.
3. p-Si is not electrically connected to anywhere. It looks electrically floated. The potential cannot be determined in this experiment. Equivalent circuit model is better to understand.
4. How to estimate the strain is not described.
5. Device structure (dimensions) is not clear.
6. What happens if compressive stress is applied.

Reviewer #3 (Remarks to the Author):

This paper claims to achieve extraordinarily high gauge factors in a semiconductor heterojunction by optoelectronic coupling. The material system is p-type 3C-SiC on p-type Si, and the gauge factors were found to be as high as 58,000, which is 2000 times that measured in the same material system without use of optoelectronic coupling. The authors claim that this is the highest ever reported for a semiconductor material, and to the best of my knowledge, this claim is true. I find this study to be novel, of significant interest to both the sensor and semiconductor materials communities and potentially of high impact. Overall I support the publication of this paper in Nature Communications. The following comments are meant to improve the quality of the manuscript and thus should be considered by the authors prior to final publication.

1. The authors claim that the approach detailed in the paper can be used to increase the gauge factor without a theoretical limit. While I agree that the data indicate that the gauge factor can be increased by a tremendous amount using the technique, it seems a bit sensational to claim "towards infinite piezoresistance" from this study. I think such a claim is a distraction from the true achievement which is orders of magnitude increase in the gauge factor. I feel that a claim "towards infinite piezoresistance" should be dropped from the paper.

2. The authors refer to the transition (depletion) region several times in the paper but they dont indicate whether this region is primarily comprised of Si or 3C-SiC. I assume that the authors believe that the depletion region is comprised primarily of Si because it is stated on page 6 that "Owing to the visible-blind property of SiC, photons are only absorbed in the depletion region and the Si layer....." It would be helpful to clarify that the depletion region extends primarily

into the Si layer based on the doping levels in each region. The cross sectional schematics suggest that the metallurgical interface is in the middle of the depletion region.

3. SAED and cross sectional TEM images of the 3C-SiC film are provided as supplementary material and show that their film has defects that are characteristic of 3C-SiC films heteroepitaxially grown on Si. It can be seen from the TEM image that the defect density is highest near the 3C-SiC/Si interface. In fact, it could be said that the 3C-SiC film consists of two distinct regions, one of high defect density and one with relatively low defect density. The density of defects at the interface is likely too high to quantify from this image. The authors did not mention anything about the defects in 3C-SiC and what role they may play in affecting the enhanced piezoresistivity. This issue is worth mentioning because the enhancement occurs in the interfacial region but is manifest on the 3C-SiC side. It is possible that the defect density ultimately puts a limit on the enhanced piezoresistivity based on charge carrier interactions with defects, etc. Defects cannot be completely eliminated in 3C-SiC grown on Si due to the lattice mismatch. This reviewer speculates that crystal quality will ultimately put a limit on the enhanced piezoresistivity, providing another reason to consider dropping "towards infinite piezoresistance" from the paper.

Response to Reviewers

We thank all reviewers for their encouraging and valuable comments, which helped significantly improve the quality of the manuscripts. The replies to all comments are provided below in blue colours, and all changes in the revised manuscript have been highlighted in yellow background.

Reviewers' comments:

Reviewer #1 (Remarks to the Author):

The paper titled, "Towards infinite piezoresistance by optoelectronic coupling" reports the modulation of the piezoresistive effect by the use of light shining on the device thereby modulating the piezoresistance. It is a nice and clear paper that shows potential for ultra-sensitive strain sensors. Adequate information has been provided to permit a clear understanding of the phenomena and the methodology involved. However, the choice of the word, "infinite" in the title is a bit puzzling and needs some explanation.

We appreciate the encouraging comments from the reviewer.

We agree with the reviewer's comment. We replaced word "infinite" by the word "giant" and whole manuscript has been revised to reflect this. The title has changed to "Giant piezoresistive effect by optoelectronic coupling in heterojunction"

It would be important to include a reference to Smith's 1954 paper. After mentioning that Smith discovered piezo-resistance in 1954, there is no reference to this seminal work. Please add a reference to the paper by Smith from 1954.

As suggested by the reviewer, we have added the reference to Smith's 1954 paper (page 1-1st paragraph 1st line and reference list)

There are numerous errors of grammar. These need to be corrected in a paper submitted to a journal with this impact.

e.g. Abstract 2nd line - the word "giantly" is out of place. It should be replaced by "greatly" or some other alternate.

page 1 - 4th paragraph 16th line - "An enhancement of up to 76% has been revealed in solar cells"

There are numerous such errors all throughout the paper that must be corrected.

We thank the reviewer for the comments.

Grammatical errors were carefully corrected throughout the paper.

Does the doping concentration, crystalline quality or defect density of the grown SiC layer influence the gauge factor?

We thank the reviewer for the interesting questions. Concentration, crystalline quality or defect density are amongst the main factors affecting the gauge factor as explained below.

Doping concentration dependence:

It has been reported that the piezoresistive coefficient decreases with doping concentration [1-6]. Because the top valence band of p-type 3C-SiC consists of three sub-bands similar to the case of p-type Si, the piezoresistive effect in p-type 3C-SiC can be qualitatively explained using the model of single crystal Si. The difference between 3C-SiC and p-type Si band structures is the distance between the spin-orbit split-off band and the heavy/light holes. The conductivity of p-type 3C-SiC is given by [7]

$$\sigma = q^2 \tau \left(\frac{P_1}{m_1} + \frac{P_2}{m_2} + \frac{P_3}{m_3} \right) \quad (1)$$

where q , P , τ , and m are electron charge, holes concentration, relaxation time, and hole effective mass, respectively. Subscripts 1, 2, and 3 refer to the heavy hole, light hole and spin-orbit split-off bands, respectively.

Applying a mechanical strain could deform and shift the top valence bands in p-type 3C-SiC [8, 9]. The shift of the sub-bands will lead to the redistribution of the hole amongst the sub-bands following the Boltzmann distribution [10, 11]. Additionally, the split and deformation of the top valence sub-bands will result in the change in effective mass in these bands. Therefore, applying a strain is expected to significantly change the electrical conductivity of p-type 3C-SiC:

$$\frac{\Delta\sigma}{\sigma} = \frac{\sum_{i=1}^3 \left[\Delta P_i \frac{1}{m_i} + P_i \Delta \left(\frac{1}{m_i} \right) \right]}{\sum_{i=1}^3 \left(P_i \frac{1}{m_i} \right)} \quad (2)$$

where ΔP_i , $\Delta \left(\frac{1}{m_i} \right)$, and $\frac{\Delta\sigma}{\sigma}$ are the hole concentration change, hole effective mass shift, and fractional change in conductivity under applied strain.

As can be seen from equation (2), the change in conductivity is inversely proportional to the carrier concentration, i.e. the piezoresistive coefficient and GF decrease with increasing the impurity concentration.

However, controlling the doping concentration of p-type 3C-SiC is a technical challenge. Experimental investigation of GF dependence on doping concentrations of p-type 3C-SiC could be an interesting topic for future development of 3C-SiC strain sensors.

Defect density influence:

As 3C-SiC thin films are grown on Si by low pressure chemical vapour deposition (LPCVD) method, the defect density is highest near the 3C-SiC/Si interface. The crystal quality of the SiC layer increases with increasing the film thickness. It was demonstrated that defect density significantly influenced the piezoresistive effect [3, 12, 13]. For example, the gauge factor turns from negative values to positive values with increasing the defect density in nanocrystalline materials [12, 14]. However, the influence of defect on the piezoresistive effect of 3C-SiC on Si substantially decrease as the thickness of 3C-SiC layer increase [3, 12, 15]. We found that with a thickness above 280 nm, the piezoresistive effect is stable and less dependent on the defect [15]. The thickness of 3C-SiC film in our experiment is 300 nm, hence we expect that the influence of defect is negligible.

What influence, if any, does the wavelength of the shining light have on the data?

Figure R1. Dependence of photovoltage on the light wavelengths. Three different wavelengths (405 nm, 522 nm, and 637 nm) with the same spot size and intensity were injected to the same position on the test sample. The lateral photovoltage increases as the wavelength increases.

We conducted additional experiments to investigate the dependence of the lateral photovoltage on the light wavelengths. We used three different light wavelengths of 405 nm, 522 nm, and 637 nm with the same intensity of 0.9 mW/cm^2 and the same spot size. As shown in Figure R1 the generated lateral photovoltage increased from approximately 14 mV to 16 mV and 20 mV for the three wavelengths 405 nm, 522 nm, and 637 nm, respectively.

The lateral photovoltage increased as the light wavelength increased, so we expect that the light wavelength could have influences on the data. However, this needs to confirm by a comprehensive investigation on effects of light condition such as light spot size, light angle, wavelength, light power. Experimental investigation of GF dependence on light parameters could be an interesting topic for our future works.

How reproducible is the data? How many cantilevers have been studied and does the data given in the paper reflect best values or average values for various cantilevers?

We used five cantilevers with the same geometry, fabricated from the same wafer and the same fabrication process. Then five cantilevers were tested in the same conditions and the same procedure. The shown data was averagely analysed from the data of five experiments.

This content has been detailly added to the paragraph 3 (line 1 to 3) page 8 in the revised manuscript, and Figure 2-a and Figure 2-b page 3 in the revised manuscript were added error bars

Figure 2: Enhancement of the piezoresistive effect. An unprecedented high gauge factor (GF) was achieved by simultaneously utilizing the lateral photovoltage and tuning the electric current to modulate the performance of the piezoresistive effect under tensile strain (a) and compressive strain (b)

Kindly address these issues as well as revise the whole paper to correct the grammatical mistakes,

The manuscript has been carefully reviewed and grammatical mistakes have been corrected as suggested by the reviewer

Reviewer #2 (Remarks to the Author):

This paper reports the photo-assisted enhancement of piezoresistive effect based on a SiC-Si heterostructure. Reported gauge factor of 58000 is highest value among reported values, but some details are missing, as pointed out below.

Q1. Meaning of V_0 is not clear. How to evaluate the resistance is not described.

V_0 is the voltage measured between the two electrodes under strain-free conditions.

The strain-free resistance R_0 is calculated by $R_0 = V_0/I$, where I is the supplied tuning current flowing between the two electrodes. This current I was kept constant throughout the measurement.

When a strain/stress is applied, the resistance will change due to the piezoresistive effect. The value of resistance is calculated by $R = V/I$, where V is the voltage measured between two electrodes under stress/strain application.

We have added the discussion on V_0 and R evaluation to the revised manuscript in paragraph 4 page 2 (line 3 to 13)

Q2. Actual piezoresistive resistance changes in application of strain is not described. In this experiment, both tuning current and photocurrent are flowing in the resistive element. Observed voltage across the resistor can be very small if the tuning current and photocurrent are almost balanced. In that situation, the large gauge factors are possibly observed. Thus, the large gauge factor seems to be a kind of trick. Actual resistance variation seems to be very small, than observed one. How to observe the actual resistance should be clear, and the resistance variation should be plotted.

We agree with Reviewer's comments at some degree. Our novel technique is to generate charge carriers from silicon substrate using photon excitation effect, and using external electric field to manipulate this photo current to achieve extremely small initial resistance R_0 .

The piezoresistive effect or the gauge factor is calculated by $GF = \frac{\Delta R}{R_0} \frac{1}{\epsilon}$, where the fractional resistance change is calculated as follow:

$$\frac{\Delta R}{R_0} = \frac{R - R_0}{R_0} = \frac{\frac{V}{I} - \frac{V_0}{I}}{\frac{V_0}{I}} = \frac{V - V_0}{V_0} = \frac{\Delta V}{V_0}$$

where, R_0 and V_0 are resistance and voltage between the two electrodes under strain-free condition.

Apparently, the fractional change in the resistance observed was extremely large as shown in Figure 2 (page 3 the revised manuscript) and Supplementary Figure 1 (page 2 the revised Supplementary Information).

As suggestion of the reviewer, we have presented the resistance variation instead of voltage variation in the revised manuscript.

Q3. *p*-Si is not electrically connected to anywhere. It looks electrically floated. The potential cannot be determined in this experiment. Equivalent circuit model is better to understand.

In our research, the *p*-Si substrate and heterojunction play critical roles in the generation and redistribution of charge carriers (electron/hole pair) into the 3C-SiC thin film. The sensing element is the SiC thin film resistor defined by two electrodes. The equivalent circuit model is shown in Figure S10 (in the revised Supplementary Information). The diode configuration of the heavily doped *p*-type 3C-SiC/*p*-type Si heterojunction (as shown in the figure) only allows the charge carriers to move from the Si side to SiC whenever there are excessive charge carriers in Si (e.g., by photon excitation). Therefore, this heterojunction configuration works well either when Si substrate is floated or kept at a potential lower than the potential on SiC side to maintain the reverse-biased condition. To confirm the later condition, we have grounded the bottom Si substrate and the results were similar to the case when the Si substrate was floated.

This content added to Supplementary Note 2 and Supplementary Figure 10 in the revised Supplementary Information.

Supplementary Figure 10: The circuit model of the sample.

Q4. How to estimate the strain is not described.

The strain application and estimation method were added to the revised Supplementary Information (Supplementary Note 1, Supplementary Figure 9, and Supplementary Table 1). The strain was calculated as below:

Supplementary Figure 9: A cantilever with load at the free end

Supplementary Figure 9 depicts a cantilever with one end clamped and one end free. The width and the thickness of the cantilever are w and t , respectively. The distance from the free end (load point) to the centre of the piezoresistor is L . t_{Si} and t_{SiC} are the thickness of the Si substrate and the SiC thin film, respectively. E_{Si} and E_{SiC} are Young's moduli of Si and SiC in the [110] orientation. A force F is applied to the free end of the cantilever.

The strain ε was calculated by using the bending model of a bi-layer beam as the SiC was epitaxially grown on the Si substrate with assumption that the bonding between the Si substrate and SiC layer is perfect. As the lengths of Si substrate and SiC layer are equal, the lateral strain of the piezoresistor is [16]

$$\varepsilon = \frac{F}{wD} L t_n = \frac{F}{wD} L \frac{t}{2} \quad (3)$$

where t_n is the distance from neutral axis to piezoresistor. The bending modulus per unit is estimated as

$$D = \frac{E_{Si}^2 t_{Si}^4 + E_{SiC}^2 t_{SiC}^4 + 2E_{Si} E_{SiC} t_{Si} t_{SiC} (2t_{Si}^2 + 3t_{Si} t_{SiC} + 2t_{SiC}^2)}{12(E_{Si} t_{Si} + E_{SiC} t_{SiC})} \quad (4)$$

Substitute the given parameters into equations (3) and (4), we can find strain at the sensing element corresponding with three different loads of 50 g, 100 g, and 150 g as shown in Table S1. We have also confirmed these results using finite element analysis (FEA) method.

Supplementary Table 1: Strain calculation results

Load (g)	F (mN)	w (mm)	t (mm)	L (mm)	E _{Si} (GPa)	E _{SiC} (GPa)	ε (ppm)
50	491	5	0.63	25	170	330	225
100	982	5	0.63	25	170	330	451
150	1473	5	0.63	25	170	330	677

Q5. Device structure (dimensions) is not clear.

We appreciate the reviewer's comment. The device structure is described clearer in the revised Supplementary Information (Supplementary Figure 4)

The length, width, and thickness of the cantilever are respectively 32 mm, 5 mm, and 0.63 mm, and the thickness of the SiC layer is 300 nm. The distance from the free end of the cantilever to the center of piezoresistor is 25 mm. The dimension of the piezoresistor is 0.5 mm x 2.5 mm, while that of the electrodes is 0.8 mm x 2.5 mm.

Supplementary Figure 4: Geometry of the cantilevers. The distance from the piezoresistor to the free end of the cantilever was 25 mm, and the whole length of the cantilever was 32 mm. The width and thickness of the cantilever were respectively 5 mm and 0.63 mm, respectively.

The thickness of the SiC layer is 300 nm. The dimensions of the piezoresistor are 0.5 mm x 2.5 mm

Q6. What happens if compressive stress is applied.

We appreciated the reviewer for the suggestion. We have conducted experiments with compressive stress as well.

Figure 2: Enhancement of the piezoresistive effect. (B) An unprecedented high gauge factor (GF) was achieved by simultaneously utilizing the lateral photovoltage and tuning the electric current to modulate the performance of the piezoresistive effect under compressive strain. **(D)** Repeatability of the fractional change in resistance under dark and light conditions as the 451 ppm compressive strain was periodically turned ON and OFF.

Supplementary Figure 1-b: The repeatability of the fractional change in resistance under light and dark conditions as different compressive strains were periodically applied (i.e., Load ON) and released (i.e., Load OFF) in the cantilevers.

As shown in Figure 2-b the fractional change in resistance $\Delta R/R_0$ linearly increased with the increase in the compressive strain, which is desirable for high-performance strain sensing applications. Figure 2-d compares the fractional change in the resistance $\Delta R/R_0$ between dark and light conditions under 451 ppm compressive strain. Under 451 ppm compressive strain,

the $\Delta R/R_0$ value increased approximately 3,000 times from -0.0087 to -27 under light condition. The resultant GF of approximately -58,000 was found under light illumination compared to the GF of about -19 under the absence of the light. Additionally, under compressive strain, the change in the resistance or GF was similar to that observed under tensile strain but opposite in sign.

The repeatability under other compressive strains were confirmed as depicted in Supplementary Figure 1-b

These results added to the revised manuscript in paragraph 4 page 2 (line 23-27, 35-38) and Figure 2-b, 2-d page 3; and Supplementary Figure S1-b in page 2 the revised Supplementary Information.

Reviewer #3 (Remarks to the Author):

This paper claims to achieve extraordinarily high gauge factors in a semiconductor heterojunction by optoelectronic coupling. The material system is p-type 3C-SiC on p-type Si, and the gauge factors were found to be as high as 58,000, which is 2000 times that measured in the same material system without use of optoelectronic coupling. The authors claim that this is the highest ever reported for a semiconductor material, and to the best of my knowledge, this claim is true. I find this study to be novel, of significant interest to both the sensor and semiconductor materials communities and potentially of high impact. Overall, I support the publication of this paper in Nature Communications. The following comments are meant to improve the quality of the manuscript and thus should be considered by the authors prior to final publication.

Q1. The authors claim that the approach detailed in the paper can be used to increase the gauge factor without a theoretical limit. While I agree that the data indicate that the gauge factor can be increased by a tremendous amount using the technique, it seems a bit sensational to claim "towards infinite piezoresistance" from this study. I think such a claim is a distraction from the true achievement which is orders of magnitude increase in the gauge factor. I feel that a claim "towards infinite piezoresistance" should be dropped from the paper

We agree with the reviewer's opinion. We replaced the word "infinite" from title and throughout the revised manuscript by the word "giant".

Q2. The authors refer to the transition (depletion) region several times in the paper but they don't indicate whether this region is primarily comprised of Si or 3C-SiC. I assume that the authors believe that the depletion region is comprised primarily of Si because it is stated on page 6 that "Owing to the visible-blind property of SiC, photons are only absorbed in the depletion region and the Si layer.... " It would be helpful to clarify that the depletion region extends primarily into the Si layer based on the doping levels in each region. The cross sectional schematics suggest that the metallurgical interface is in the middle of the depletion region.

We thank reviewer for point this out. We have revised the manuscript as follows:

Figure 5-a page 5 in the revised manuscript was revised to show that the depletion region extends primarily into the Si layer. The metallurgical interface is primary in the Si layer side.

Figure 5-a: Electron-hole pair generation and separation. Under visible light, electron-hole pairs (EHPs) are generated in the depletion region and Si substrate. Then, the generated EHPs in the depletion region are separated by an internal electric field, and the generated holes in Si substrate tunnel to the 3C-SiC thin film.

We also added the content “It is worth noting that the depletion region extends primarily into Si substrate (Figure 5-a) because the carrier concentration in the Si substrate ($5 \times 10^{14} \text{ cm}^{-3}$) is much lower than that in the SiC thin film ($5 \cdot 10^{18} \text{ cm}^{-3}$)” to the revised manuscript in paragraph 1 page 7 (line 1-5).

Q3. SAED and cross sectional TEM images of the 3C-SiC film are provided as supplementary material and show that their film has defects that are characteristic of 3C-SiC films heteroepitaxially grown on Si. It can be seen from the TEM image that the defect density is highest near the 3C-SiC/Si interface. In fact, it could be said that the 3C-SiC film consists of two distinct regions, one of high defect density and one with relatively low defect density. The density of defects at the interface is likely too high to quantify from this image. The authors did not mention anything about the defects in 3C-SiC and what role they may play in affecting the enhanced piezoresistivity. This issue is worth mentioning because the enhancement occurs in the interfacial region but is manifest on the 3C-SiC side. It is possible that the defect density ultimately puts a limit on the enhanced piezoresistivity based on charge carrier interactions with defects, etc. Defects cannot be completely eliminated in 3C-SiC grown on Si due to the lattice mismatch. This reviewer speculates that crystal quality will ultimately put a limit on the enhanced piezoresistivity, providing another reason to consider dropping "towards infinite piezoresistance" from the paper.

We appreciate the reviewer's comments.

We agree that the lateral photovoltage might be affected by crystalline quality or defect density near the 3C-SiC/Si interface, hence probably affecting the GF. As can be seen from the TEM image (Supplementary Figure 3-c) that the defect density is highest near the 3C-SiC/Si interface up to 65 nm from the interface, and the quality of the SiC thin film increases with increasing thickness of SiC film. Therefore, when the thickness of the SiC thin film increases, the effect of defect density decreases. We found that with a thickness of SiC layer above 280 nm, the piezoresistive effect in dark condition was independent on the defect density. The

thickness of SiC film in our device is 300 nm, hence we expect that effect of defect in heterojunction area is negligible under light condition.

We have dropped the word “finite” in the tile and in the revised manuscript.

Reference for response to the reviewers

- [1] O. Tufte and E. Stelzer, "Piezoresistive properties of silicon diffused layers," *Journal of applied physics*, vol. 34, no. 2, pp. 313-318, 1963.
- [2] O. Tufte and E. Stelzer, "Piezoresistive properties of heavily doped n-type silicon," *Physical review*, vol. 133, no. 6A, p. A1705, 1964.
- [3] M. Eickhoff and M. Stutzmann, "Influence of crystal defects on the piezoresistive properties of 3C-SiC," *Journal of applied physics*, vol. 96, no. 5, pp. 2878-2888, 2004.
- [4] Y. Kanda, "A graphical representation of the piezoresistance coefficients in silicon," *IEEE Transactions on electron devices*, vol. 29, no. 1, pp. 64-70, 1982.
- [5] J. S. Shor, D. Goldstein, and A. D. Kurtz, "Characterization of n-type beta-SiC as a piezoresistor," *IEEE transactions on electron devices*, vol. 40, no. 6, pp. 1093-1099, 1993.
- [6] T. Toriyama and S. Sugiyama, "Analysis of piezoresistance in p-type silicon for mechanical sensors," *Journal of microelectromechanical systems*, vol. 11, no. 5, pp. 598-604, 2002.
- [7] C. Kong, W. Wang, K. Liao, Y. Ma, S. Wang, and L. Fang, "The theoretical studies of piezoresistive effect in diamond films," *Science in China Series A: Mathematics*, vol. 45, no. 1, pp. 107-114, 2002.
- [8] L. Wenchang, Z. Kaiming, and X. Xide, "Strain effects on the band structures of beta-SiC," *Journal of Physics: Condensed Matter*, vol. 5, no. 7, p. 883, 1993.
- [9] R. Rahimi, C. Miller, S. Raghavan, C. Stinespring, and D. Korakakis, "Electrical properties of strained nano-thin 3C-SiC/Si heterostructures," *Journal of Physics D: Applied Physics*, vol. 42, no. 5, p. 055108, 2009.
- [10] J. Richter, J. Pedersen, M. Brandbyge, E. V. Thomsen, and O. Hansen, "Piezoresistance in p-type silicon revisited," *Journal of Applied Physics*, vol. 104, no. 2, p. 023715, 2008.
- [11] G. L. Bir and G. E. Pikus, "Symmetry and strain-induced effects in semiconductors," 1974.
- [12] M. Eickhoff, M. Möller, G. Kroetz, and M. Stutzmann, "Piezoresistive properties of single crystalline, polycrystalline, and nanocrystalline n-type 3 C-SiC," *Journal of applied physics*, vol. 96, no. 5, pp. 2872-2877, 2004.
- [13] X. Song *et al.*, "Evidence of electrical activity of extended defects in 3C-SiC grown on Si," *Applied Physics Letters*, vol. 96, no. 14, p. 142104, 2010.
- [14] M. Eickhoff, H. Möller, J. Stoemenos, S. Zappe, G. Kroetz, and M. Stutzmann, "Influence of crystal quality on the electronic properties of n-type 3C-SiC grown by low temperature low pressure chemical vapor deposition," *Journal of applied physics*, vol. 95, no. 12, pp. 7908-7917, 2004.
- [15] H.-P. Phan *et al.*, "Thickness dependence of the piezoresistive effect in p-type single crystalline 3C-SiC nanothin films," *Journal of Materials Chemistry C*, vol. 2, no. 35, pp. 7176-7179, 2014.
- [16] X. Gao, W.-H. Shih, and W. Y. Shih, "Induced voltage of piezoelectric unimorph cantilevers of different nonpiezoelectric/piezoelectric length ratios," *Smart Materials and Structures*, vol. 18, no. 12, p. 125018, 2009.

Reviewers' comments:

Reviewer #1 (Remarks to the Author):

The revised manuscript and the response to the reviewers are quite satisfactory from my perspective. I still feel that enough attention has not been paid to the defects in the 3C-SiC. But given that it is rather difficult to correlate specific defect densities with measured GFs (especially since the defects are quite complex and interact with each other particularly under strain), I would agree that the present explanation is adequate. A more detailed study should be attempted if possible to look at defect densities and their influence.

On page 8 of your manuscript, in the last line of text, please correct the instrument name to read "Keithley 2450 SourceMeter" and not "Keithley 2450 Source Metre"

Reviewer #2 (Remarks to the Author):

I understand most of parts, but I still cannot understand this point.

1. According to your reply, the gauge factor is defined as
 $\Delta R/R_0 = (V - V_0)/V_0$.

If the photovoltage V_p is generated, above equation becomes
 $\Delta R/R_0 = (V - V_0)/(V_0 + V_p)$.

If the value V_0 is close to $-V_p$, the gauge factor reaches an infinite value in calculation.

Even if the gauge factor seems to be enhanced, SN ratio might not be improved. Did you observed the SN ratio of strain detection by photo illumination.

Reviewer #3 (Remarks to the Author):

In my opinion, the authors have sufficiently addressed all concerns raised in my original review, as well as the concerns raised by the other reviewers, to merit publication with the exception of one small but significant issue related to the use of the word "infinite" in reference to the piezoresistive effect in the structures described in the paper. The authors eliminated the use of the word "infinite" in the revised manuscript in all locations except one. The following was extracted directly from the abstract:

"This value can theoretically be infinitely increased".

Like I stated in my initial review, it seems a bit sensational to claim that the piezoresistive effect, and resulting gauge factor, can be infinitely increased. And I question that theory would support this claim. It is my opinion that this sentence be taken out of the abstract as a condition for publication.

Response to Reviewers

We thank all reviewers for their encouraging and valuable comments, which helped significantly improve the quality of the manuscripts. The replies to all comments are provided below in blue colours, and all changes in the revised manuscript have been highlighted in yellow background.

Reviewers' comments:

Reviewer #1 (Remarks to the Author):

The revised manuscript and the response to the reviewers are quite satisfactory from my perspective. I still feel that enough attention has not been paid to the defects in the 3C-SiC. But given that it is rather difficult to correlate specific defect densities with measured GFs (especially since the defects are quite complex and interact with each other particularly under strain), I would agree that the present explanation is adequate. A more detailed study should be attempted if possible to look at defect densities and their influence.

On page 8 of your manuscript, in the last line of text, please correct the instrument name to read "Keithley 2450 SourceMeter" and not "Keithley 2450 Source Metre"

We thank the reviewer for the comments.

As suggested by the reviewer, we have corrected the phrase "Keithley 2450 Source Metre" as "Keithley 2450 SourceMeter" in the revised manuscript in paragraph 3 page 8 (line 17 to 18).

Reviewer #2 (Remarks to the Author):

I understand most of parts, but I still cannot understand this point.

1. According to your reply, the gauge factor is defined as

$$\Delta R/R_0 = (V - V_0)/V_0.$$

If the photovoltage V_p is generated, above equation becomes

$$\Delta R/R_0 = (V - V_0)/(V_0 + V_p).$$

If the value V_0 is close to $-V_p$, the gauge factor reaches an infinite value in calculation.

Even if the gauge factor seems to be enhanced, SN ratio might not be improved. Did you observed the SN ratio of strain detection by photo illumination.

We thank the reviewer for the comments.

1. The calculation of the gauge factor (GF) is explained clearer as follow:

To estimate the gauge factor under the light condition, the chip is firstly illuminated. Then, the tuning current I is supplied. Next, the initial voltage V_0 is measured under strain-free condition. Therefore, the photovoltage V_p is already included in V_0 . Finally, load is applied and the voltage V is measured. The calculation of GF can be presented as below:

$$GF = \frac{\Delta R}{R_0} \frac{1}{\varepsilon}$$

where R_0 is strain-free resistance under light illumination, ε is the applied strain, ΔR is the resistance change due to applied strain: $\Delta R = R - R_0$. R_0 and R are calculated by

$$R_0 = \frac{V_0}{I} \text{ and } R = \frac{V}{I}$$

Note, the tuning current I is kept constant throughout the experiment.

The GF is therefore calculated by

$$GF = \frac{\Delta R}{R_0} \frac{1}{\varepsilon} = \frac{R - R_0}{R_0} \frac{1}{\varepsilon} = \frac{\frac{V}{I} - \frac{V_0}{I}}{\frac{V_0}{I}} \frac{1}{\varepsilon} = \frac{V - V_0}{V_0} \frac{1}{\varepsilon} = \frac{\Delta V}{V_0} \frac{1}{\varepsilon}$$

2. The signal to noise ratio (SNR) is calculated by:

$$SNR = 20 \log_{10} \frac{V}{V_n}$$

where V is the signal voltage and V_n is the noise voltage. We observed that SNR was 70 dB under illumination and 30 dB under the dark condition. It can be seen that the SNR was increased significantly under illumination.

We add the content “In addition, the signal to noise ratio (SNR) under light condition increased significantly in comparison with that under dark condition” to the revised manuscript in paragraph 1 page 3 (line 2 to 4).

Reviewer #3 (Remarks to the Author):

In my opinion, the authors have sufficiently addressed all concerns raised in my original review, as well as the concerns raised by the other reviewers, to merit publication with the exception of one small but significant issue related to the use of the word "infinite" in reference to the piezoresistive effect in the structures described in the paper. The authors eliminated the use of the word "infinite" in the revised manuscript in all locations except one. The following was extracted directly from the abstract:

"This value can theoretically be infinitely increased".

Like I stated in my initial review, it seems a bit sensational to claim that the piezoresistive effect, and resulting gauge factor, can be infinitely increased. And I question that theory would support this claim. It is my opinion that this sentence be taken out of the abstract as a condition for publication.

We appreciate the reviewer's comments.

As suggested by the reviewer, the sentence "This value can theoretically be infinitely increased" has been removed from the abstract.

Reviewers' comments:

Reviewer #1 (Remarks to the Author):

The concerns raised seemed to have been addressed.

Reviewer #2 (Remarks to the Author):

This paper is much improved, I think the paper is acceptable to publish.

Reviewer #3 (Remarks to the Author):

All issues by the reviewers have been adequately addressed. The manuscript is now suitable for publication.